## Perspective

microbiology, health and disease and epidemiology, cellular biology

parasite, co-infection, trypanosoma, plasmodium

**Author for correspondence:**
Eleanor Silvester
e-mail: es461@cam.ac.uk

# Parasite co-infection: an ecological, molecular and experimental perspective

Frank Venter[1], Keith R. Matthews[1] and Eleanor Silvester[1,2]

[1]Institute for Immunology and Infection Research, Ashworth Laboratories, School of Biological Sciences, University of Edinburgh, Scotland EH9 3FL, UK
[2]Cambridge Institute for Medical Research, University of Cambridge, Cambridge Biomedical Campus, The Keith Peters Building, Hills Road, Cambridge CB2 0XY, UK

FV, 0000-0002-1925-2510; KRM, 0000-0003-0309-9184; ES, 0000-0001-9827-1412

Laboratory studies of pathogens aim to limit complexity in order to disentangle the important parameters contributing to an infection. However, pathogens rarely exist in isolation, and hosts may sustain co-infections with multiple disease agents. These interact with each other and with the host immune system dynamically, with disease outcomes affected by the composition of the community of infecting pathogens, their order of colonization, competition for niches and nutrients, and immune modulation. While pathogen-immune interactions have been detailed elsewhere, here we examine the use of ecological and experimental studies of trypanosome and malaria infections to discuss the interactions between pathogens in mammal hosts and arthropod vectors, including recently developed laboratory models for co-infection. The implications of pathogen co-infection for disease therapy are also discussed.

## 1. Introduction

The aim of scientific exploration in the laboratory is to control as many factors as possible, allowing the understanding of the impact of only one, or a few, variables. This has underpinned almost all pathogen research and generated impressive insight into the molecular mechanisms of pathogen invasion, establishment and immune evasion, as well as the response of the host defences. However, in almost all cases, such studies cannot represent the context and setting of pathogen infections in the real world. In particular, pathogens rarely exist in isolation but rather enter hosts already harbouring a diversity of pre-existing commensal or pathogenic organisms, each imposing their own effects on the within-host environment and immune system [1–3]. Furthermore, the community of organisms within a host may be unstable or destabilized by the new ingression. As a consequence, a host represents a complex and dynamic environment in which pathogens must establish and optimize their survival in competition with co-infecting organisms while also avoiding the defences of the host [4]—which themselves may be modulated by the pre-existing congregation. This introduces considerable complexity at any given time and also dynamically over time. This complexity is compounded where pathogens are transmitted by disease vectors, where the contribution of the vector's biology and epidemiology, as well as its own community of microbial passengers, generates additional interactions, potentially further influenced by the impact of environmental change [5]. These combined interactions, which are so important in the real world, cannot be fully disentangled using laboratory studies alone. In this review, we discuss the different approaches that can be used to understand the impact of co-infections, with a focus on two microbial eukaryotic pathogens, causing malaria (*Plasmodium* spp.; transmitted by mosquitoes) [6] and African trypanosomiasis (*Trypanosoma brucei*, *Trypanosoma congolense* and *Trypanosoma vivax*; transmitted by tsetse flies) [7,8]. Focusing on these parasites, we review diverse studies on the epidemiology of co-infection, and

**Figure 1.** A hypothetical scenario which illustrates some of the factors that shape the composition of a microbial community. These include (i) direct and indirect interactions between co-infecting microorganisms; (ii) the availability of vectors and their capacity to transmit different parasite species or strains; (iii) nutrition status and (iv) host genetics which impact upon host immune responses and susceptibility to infection; (v) environmental factors such as soil conditions and climate. Created with BioRender.com. (Online version in colour.)

the factors that are important to consider when using experimental systems to understand how these organisms sustain themselves in their mammalian host in the context of co-infection, extending beyond the elegant immunological and molecular studies focused on immune evasion by individual infections in host model systems. The mechanisms through which trypanosomes and *Plasmodium* spp. detect and respond to co-infecting parasites are then discussed. Finally, we highlight how these interactions can be best understood through an interdisciplinary approach involving a combination of infection biology, epidemiology, mathematical modelling and evolutionary theory (figure 1).

## 2. Epidemiology of co-infection

Most studies of co-infection have focused on the analysis of hosts and pathogens at the epidemiological level. Hosts in natural systems are frequently infected with a diverse community of microorganisms composed of different taxa, different species or even strains of the same organism. This can be illustrated by a recent longitudinal study from western Kenya, where a cohort of 548 zebu calves were found to be infected with over 50 different pathogens, including many trypanosome and apicomplexan parasites [9]. With respect to trypanosomes, the potential for infection with multiple species is particularly high because these parasites have an unusually broad host range, allowing frequent interactions between species in diverse hosts. Reflecting this, a large number of studies in different regions have reported the co-circulation of *T. brucei*, *T. congolense* and *T. vivax*, and these have been detected both in surveyed livestock and game animals [10], as well as in trapped tsetse flies (e.g. [11,12]). The major human infective *Plasmodium* species, *Plasmodium falciparum*, in contrast shows strict host specificity although other human infective malaria parasites, such as *Plasmodium knowlesi*, are more promiscuous and zoonotic.

Alongside mammalian hosts that are multiply infected with different parasite species, vectors can also sustain and transmit multiple parasites. In some cases, the same vector can transmit different pathogens, as is the case for mosquito transmission of *Plasmodium* spp. and *Wuchereria bancrofti* [13]. The same mosquito species can also be infected by different species of *Plasmodium* [14], although host preference might impose some limits on the prevalence of co-infecting parasites in the vector. Similarly, multiple trypanosome species, and strains of the same species, can simultaneously infect their tsetse vector [15,16] although different subspecies of tsetse vary in their transmission of distinct trypanosome species, contributed to by their geographical distribution and anthropophilic or zoophilic preferences [17]. Climate factors can also determine the distribution of vector species [18,19] and may alter the infection dynamics of the parasites that they harbour, influencing their potential to sustain or establish co-infections. For example, tsetse flies are particularly sensitive to changes in temperature and humidity, as illustrated by the shifts in fly distribution observed in northern Zimbabwe [20,21]. Interestingly, vector availability itself does not necessarily mean high disease incidence. For example, a longitudinal cohort study of calves in western Kenya found a low incidence of clinical trypanosomiasis in an area with high tsetse challenge [9].

While epidemiological studies continue to provide insight into the complicated drivers of co-infections, important knowledge gaps remain, sometimes driven by the tractability of the pathogen or the research priorities and focus of the research community exploring the pathogen. This was highlighted in a recent meta-analysis which found that only 0.05% of co-infection studies in humans focused on helminth parasites, despite their profound global burden [22]. A consequence of this bias can be a focus in laboratory studies on species such as *T. brucei* or *P. falciparum*, at the expense of other species that are not, or less established, human pathogens (*T. vivax* and *T. congolense*; *P. knowlesi*). This has the potential to limit surveillance studies in the field to the better-studied parasites, although the advent of metagenomic rather that species-targeted analyses will alleviate this issue. It has also been highlighted that abundance studies do not consider the within-host dynamics of co-infecting organisms, reflected by the variation seen in such data between individuals in the same geographical location [23–25]. Combined, this highlights the need for more experimental and field studies to tease apart the delicate interactions between different species or strains co-infecting the same host and the value of longitudinal surveillance and chronic infection studies to monitor dynamic temporal changes in co-infection prevalence. Only with data from comprehensive and unbiased epidemiological studies can researchers in the laboratory understand what are the important interactions happening in the real world, and as a result focus their research on co-infections that have the potential to impact human and animal health most severely.

## 3. Considerations in experimental studies of co-infection

Given the challenges of experimentally manipulating co-infection in the field, experimental systems in the laboratory are particularly valuable. In recent years, both trypanosome and *Plasmodium* systems have become excellent experimental models to explore parasite co-infections in the mammalian host, providing a bridge between the complexity of field-based studies and the accessibility of laboratory studies that have traditionally focused on monoinfection. This is because, for each parasite, there are available tools to mark or identify distinct parasite strains or species and the ability to discriminate proliferative and transmission-adapted

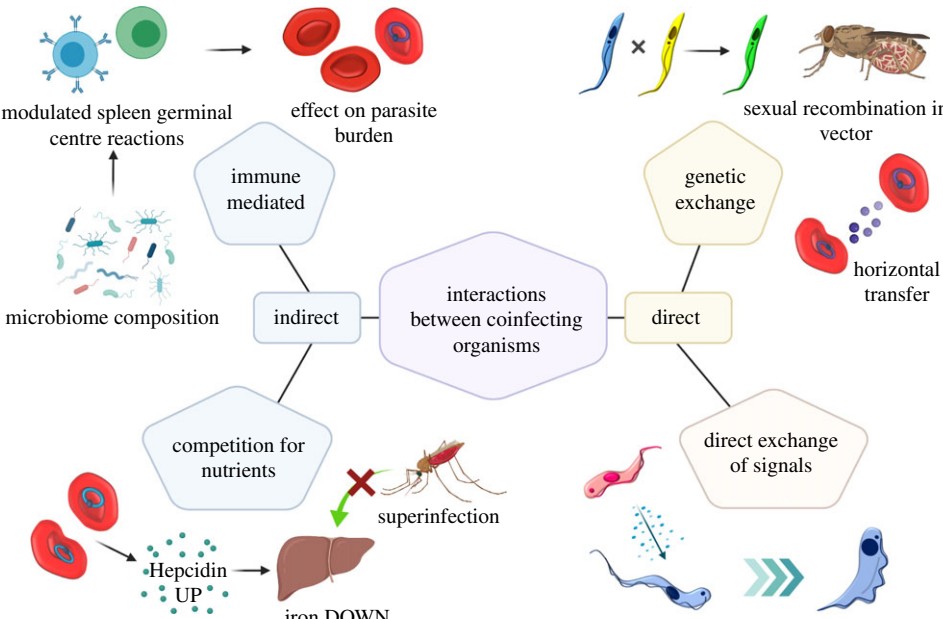

**Figure 2.** Mechanisms that can mediate the interactions between co-infecting organisms. Examples of direct and indirect mechanisms of interaction are illustrated. Genetic exchange: sexual exchange between African trypanosomes in a co-infected tsetse fly, or horizontal transfer of DNA between *Plasmodium* infected red blood cells via extracellular vesicles. Direct exchange of signals: interspecies quorum sensing between African trypanosomes altering transmission potential. Competition for nutrients: an established *Plasmodium* infection increases host production of hepcidin diverting iron away from the liver, inhibiting invasion by superinfecting parasites. Immune mediated: the microbiota can influence the immune environment of a host to affect malaria infection. Created with BioRender.com. (Online version in colour.)

forms. There are also relevant culture systems and rodent infection models. This has permitted experimental studies of co-infection, the design of which can substantially alter the outcome. Considerations include the timing and order of infection, as well as how pathogen virulence and transmission potential are modulated in multiply infected hosts. The impact of acute and chronic co-infection profiles is also relevant. In this section, we discuss the factors that can influence the outcome of experimental studies of co-infection.

## (a) Time and space

An important consideration for co-infection studies relates to when the respective parasites are introduced experimentally. In one scenario, there is infection with both pathogens at the same time, for example by a single insect bite from a co-infected vector, or through simultaneous inoculation of more than one parasite strain or species via syringe. The second scenario involves temporally offset superinfection of an already infected host by a second pathogen strain or species. Given the chronic nature of many infections, superinfection is likely to be the more common route to establishing a co-infection in the real world and evidence suggests that there is often a competitive advantage to being first on the scene. For example, goats already infected with *T. congolense* showed delayed appearance of *T. brucei* (compared to uninfected goats) after superinfection by tsetse fly bite [26]. Experimental sequential infection with *T. congolense* isolates in cattle also suppressed the prevalence of the incoming parasites [27]. On the other hand, there was no delay in superinfection by another trypanosome species, *T. vivax*, in goats already infected with *T. congolense* [26].

The importance of timing has also been shown by different studies analysing the consequence of co-infection between *T. brucei* and *Plasmodium berghei* in mice. One study revealed a heightened number of both parasites when co-infected, with more severe anaemia, hypoglycaemia and lower survival

[28]. More recently, however, an initial infection with *T. brucei* was found to limit subsequent *P. berghei* establishment in the liver, and protect mice from cerebral malaria and prolong survival [29]. In the first study, parasites were inoculated simultaneously generating enhanced *Plasmodium* virulence, whereas in the latter study *T. brucei* was established first, generating reduced virulence. Similarly, a recent study of mixed-species malaria co-infection in mice found increased virulence when *Plasmodium yoelli* was inoculated at the same time as *Plasmodium vinckei* or *Plasmodium chabaudi* [30]. The authors highlight differences between their findings and others, acknowledging that timing, route of inoculation and host strain or species can all alter the co-infection outcome. Such studies indicate that interactions which modulate virulence may occur in human malaria co-infections, but their nature remains unpredictable. Thus, it is important when designing co-infection studies to consider how different experimental designs can deliver contrasting outcomes. It may be necessary to carry out multiple permutations of any one experiment to better understand how two pathogens could influence each other in the field. Epidemiological information, for example whether two parasites commonly share a vector, can help to predict whether simultaneous or sequential inoculation better represents the field setting.

In addition to the relative timing of infection, the spatial coincidence of parasites can be important in co-infection studies where the proximal interaction of parasites might be relevant. For example, two parasites which inhabit macrophages, *Leishmania infantum* and *Toxoplasma gondii*, can reside within the same cell when inoculated on the same day [31]. However, when *T. gondii* was inoculated five days later than *L. infantum*, both parasites established infection but not within the same macrophage. Interestingly, mice infected with *L. infantum* were also protected from the usual virulence of *T. gondii*. For *Plasmodium*, the simultaneous infection of a single red blood cell can occur frequently in culture and *in vivo* using parasites

of the same species [32]. However, there is complexity in the potential for the co-occupation of erythrocytes between co-infecting species, since *P. vivax* and *P. ovale* favour young erythrocytes, *P. malariae* favours older red blood cells, whereas *P. falciparum* is quite promiscuous with respect to red blood cell age. These preferences are not absolute and the relative proportion of red blood cell types available to co-infecting parasites might be dynamic within the host. The relative success of each parasite strain or species in a *Plasmodium* co-infection could therefore vary substantially depending on the host's red blood cell landscape. This points to the value of incorporating studies focused on host biology in affected populations to build a more complete picture of co-infections. Experimental studies could benefit from testing co-infection interactions in diverse host environments to see whether observations are widespread or restricted to a given scenario.

## (b) Transmission and virulence

Monitoring the modulation of virulence is an important consideration in co-infection studies, whether virulence is defined as overall parasite number, or parasite-induced host pathology which is the focus of most of the following discussion [33,34]. A meta-analysis of co-infection data found that most studies reported negative impacts for human health [22] and co-infection can exacerbate detrimental effects caused by individual pathogens. For example, *P. falciparum* and hookworm contribute to host anaemia by distinct mechanisms and in co-infection these effects could be additive [35].

Alternatively, co-infection can result in reduced virulence where parasites with different levels of relatedness directly interfere with each other. This interference could have an associated parasite fitness cost, reducing the extent to which co-infecting parasite species are able to exploit the host [36]. For example, a survey of African trypanosomiasis in the Gambia revealed that co-infection with *T. vivax* and *T. congolense* was associated with reduced pathology compared to *T. congolense* infection alone [37], suggesting these different species could interfere to reduce virulence. However, a mixed infection comprising an avirulent strain and a virulent strain of the same species, *T. brucei*, also reduced the deleterious effects on the host relative to an infection with the virulent strain alone [38]. This might reflect that overall relatedness between co-infecting parasites is not important in determining virulence outcomes, but instead the extent of diversity or relatedness at key virulence gene loci.

As well as virulence, transmissibility can be affected by co-infections, with virulence and transmissibility often interacting [39]. Various studies have focused on the impact of co-infection on malaria transmission potential through monitoring gametocyte investment. One study featuring two *P. chabaudi* clones of differing virulence did not reveal increased investment in transmission stages by either clone, despite competitive suppression of asexual parasite density in the co-infection [40]. However, the investment in transmission in the host may change dynamically in response to parasite numbers, allowing a short-term or long-term transmission strategy in the context of signals from co-infecting *Plasmodium* strains [41]. Likewise, in studies of human malaria, co-infection with *P. malariae* was associated with increased gametocyte production by *P. falciparum* in three out of four study sites in malaria-endemic regions [42]. On the other hand, co-infection with *P. vivax* was associated with lower *P. falciparum* gametocyte density in

patients in Thailand, although it was unclear whether this was influenced by patients seeking medical attention earlier in cases of co-infection or whether there was an inhibition of *P. falciparum* gametocytogenesis [43]. Co-infection with other parasites can also affect malaria transmission. Humans simultaneously infected with helminths and *P. falciparum* presented a higher malaria transmission potential through increased gametocyte production [44]. Interactions between parasites within the vector may also influence transmission success of individual strains. For example, mosquitoes infected with one strain of *P. chabaudi* are more susceptible to infection by further strains during subsequent blood meals [45]. All of these examples highlight that the balance between within-host replication and the preparation for onward transmission can be sensitive to the presence of co-infecting parasites, of different strains or species, but that the outcome can be difficult to predict and may show plasticity in different experimental or clinical conditions. This emphasizes the need for wide-ranging and collaborative research, as transmission and virulence impacts observed in one study are unlikely to be applicable universally. With more data, the impact of co-infection and the potential outcomes of intervention can be better predicted at a local level.

Co-infection might also contribute to the trypanosome transmission potential of the tsetse for example through influencing the coordinated 'social motility' of the parasites in tsetse fly stages of the life cycle. Experimental studies of this phenomenon *in vitro* demonstrate avoidance behaviours of parasites of the same species and strain, monitored via the trajectory of growth spurs on culture plates [46]; whether a similar phenomenon occurs between species or strains has not been tested but has the potential to alter the migration of the parasites during maturation in the vector if operational in the fly. Also, the distinct swimming behaviours of different trypanosome species [47] could generate the potential for interference if different parasite species occupy the same vector simultaneously, since coordinated swimming may contribute to the parasites' journey through the fly [48]. Interestingly, colonization of tsetse fly salivary glands by *T. brucei* has also been shown to alter the anti-haemostatic activity of the saliva, impairing blood-feeding ability [49]—this promoting multiple feeding cycles and potentially increasing co-infection likelihood.

## (c) Acute versus chronic infections

Another aspect to consider when investigating experimental co-infections is dynamic interactions in acute versus chronic stages of infection, reflecting the mismatch that can be seen in longitudinal versus cross-sectional field studies. Co-infection may be missed in cross-sectional studies when one pathogen is competitively suppressing another at one point in time, but with fluctuations in their relative dominance over time. This is illustrated in a study of children with asymptomatic *Plasmodium* infection in Papua New Guinea, which monitored the dynamics of multiple *Plasmodium* species and genotypes over a 60-day period. This revealed that overall parasitaemia remained relatively stable but that there were dramatic shifts in the representation of different parasite genotypes over time [50]. These complex co-infection dynamics would have been missed in a cross-sectional study. The authors suggest that the observed dynamics may be explained by a combination of density-dependent regulation between species, antigenic

variation and immune clearance. Once a species-specific response clears the majority species the density would fall below the threshold allowing other co-infecting species to expand, leading to the sequential pattern of infection observed.

These examples all highlight that tractable experimental studies can provide important information on the outcome of co-infections. However, they also highlight that the observed outcomes can be significantly affected by the experimental design and that this should be informed by knowledge of the biological interactions between co-infecting parasites in the field, where possible.

# 4. Mechanisms of interaction that operate in co-infections

Once experimental conditions are established for monitoring co-infections, a mechanistic understanding of interactions becomes feasible. Clearly, modulation of the host immune response by pathogens has important consequences for co-infection opportunity and outcome but this topic has been thoroughly explored in other reviews [51–54]. Below we focus on the parasite-intrinsic molecular processes that parasites in co-infections use to interact, directly or indirectly, specifically through the exchange of genes and protein factors, whether as part of sexual processes, compartmentalized in vesicles or as soluble signals (figure 2).

## (a) Sexual exchange

Eukaryotic parasites typically have a capacity for sexual exchange. In *T. brucei* meiosis is not an obligatory part of the life cycle but can take place within the tsetse fly. This permits new variants of human infective trypanosome to be generated through sexual exchange between *T. b. rhodesiense* and *T. b. brucei* in a co-infected tsetse [55], providing the opportunity for the acquisition of the SRA gene, important for human infectivity, but also other alleles. This is possible because *T. brucei* and *T. b. rhodesiense* cohabit the tsetse salivary glands, where meiosis occurs. *T. b. gambiense* type 1 also occupies this niche but appears asexual, without evidence of sexual exchange within or between species; there is evidence for *T. b. gambiense* type 2 sexual exchange, but these parasites may now have become extinct [56]. Beyond the *T. brucei* group of trypanosomes, opportunities for cross species sexual exchanges are absent because each species matures at a different site in the fly: *T. brucei* spp. (salivary glands), *T. congolense* (proboscis) and *T. vivax* (mouthparts). However, there is evidence for within-species sexual exchange for *T. congolense* [57] although there is some controversy around this, with others supporting a clonal population structure [58]. By contrast to *T. brucei* and *T. congolense*, *T. vivax* does not appear to undergo mating [59].

Unlike trypanosomes, sex is essential in the life cycle of *Plasmodium* resulting from the fusion of male and female gametocytes produced in the bloodstream to form a zygote in the mosquito gut. Where mosquito infections are initiated with gametocytes of different genetic backgrounds, there are significant opportunities for parasite co-infection to have important consequences for the epidemiology of the parasite in the field, for example in the exchange and spread of drug resistance between parasite strains [60]. Interestingly, infections with a single genotype can enhance mosquito

infectivity compared with mixed genotype infections [61]. Consequently, where the component parasites differ in drug sensitivity, there is a risk that therapeutic intervention could generate enhanced disease transmission in a geographical location through the selection for monoinfections after competitive release [61–63].

## (b) Extracellular vesicles

A mechanism where co-infecting pathogens can interact directly throughout the life cycle is through the exchange of virulence factors, via either gene or protein transfer. In bacterial systems, horizontal gene transfer can lead to the transfer of antimicrobial resistance genes between strains and species, and potentially from commensals to pathogens [64]. Horizontal gene transfer may occur through direct cell-to-cell contact by bacterial conjugation or via the exchange of extracellular vesicles [65]. Extracellular vesicles have also been proposed to exchange the SRA virulence factor between subspecies of *T. brucei*, transferring resistance to human serum. In co-culture experiments, SRA was transferred to *T. brucei* via extracellular vesicles generated from shed nanotubes [66], rendering the recipient cells resistant to trypanolysis in human serum. In tsetse fly stages, extracellular vesicles can also affect social motility that may influence the coordinated migration of the parasites through the insect gut [67]. Extracellular vesicles can also mediate transfer of DNA between infected red blood cells in *P. falciparum* co-cultures resulting in the transfer of drug-resistance genes [68]. Additionally, signalling through extracellular vesicles can promote differentiation of *Plasmodium* parasites to transmissible gametocyte stages in culture [69] such that, in combination, virulence or drug-resistance genes can be exchanged between strains in a co-infection, helping new variants to be generated and transmitted. More recently extracellular vesicles containing lactate dehydrogenase derived from density-stressed *P. falciparum* cultures were found to limit the growth of low-density parasite populations *in vitro* by triggering apoptotic events, suggesting an intercellular signalling mechanism that could regulate parasite population density [70].

## (c) Soluble signalling factors

Single-celled organisms have developed soluble communication systems that enable composite members to coordinate behaviours as a community. One example is 'quorum sensing', which involves regulation of gene expression through the production and detection of signalling molecules whose concentration increases with cell density. Quorum sensing is best understood in bacteria where a variety of signalling mechanisms are known, but also plays a role in the community behaviour of diverse eukaryotic pathogens. For example, the development of *T. brucei* from a proliferative slender form to a transmission-adapted stumpy form in its mammal host is regulated by quorum sensing via 'stumpy induction factor' activity [71,72]. This has been found to comprise oligopeptide signals generated by parasite-released peptidases [73], with intracellular signalling pathway components required for quorum sensing also identified [74]. Where both the signal generation and transduction pathways are shared between co-infecting strains or species there is the potential to influence the infection outcome for the host and the pathogen. This has been observed experimentally, with *T. congolense* able to promote accelerated *T. brucei* differentiation to stumpy forms in

a co-infection through shared quorum-sensing signals [75]. *Trypanosoma congolense* also exhibits density-dependent arrest and has the machinery for quorum sensing, which can functionally complement the equivalent molecules in *T. brucei*.

## (d) Indirect interactions

Infectious organisms may also interfere with each other's growth and establishment through indirect interactions involving modification of the host environment and competition for nutrients. Iron is a valuable commodity to compete for in co-infections and microbes have developed mechanisms to increase their share, for example, the deployment of iron-scavenging siderophores by bacterial and fungal pathogens. For *Plasmodium*, modified host iron responses can allow parasites that have established a blood-stage infection to prevent superinfection by a competitor species or strain [76]. This is possible because the blood-stage *Plasmodium* infection increases host production of the iron-regulatory hormone hepcidin in a density-dependent manner. This diverts iron away from the liver, impairing the liver-stage development of newly inoculated parasites, and making the host refractory to superinfection. Environmental factors can also modulate asexual growth and the generation of transmission stages in *Plasmodium*. In particular, *P. falciparum* (but not all *Plasmodium* species) monitor the presence of Lyso-phosphatidyl choline (LyosPC) within the infected mammalian host, with LysoPC repressing gametocyte formation. Upon LyosPC depletion with elevated infection levels, gametocyte formation is promoted [77]. Consequently, differing sensitivity or competition for LysoPC could potentially generate distinct probabilities of sexual maturation between strains, favouring relative virulence or transmission potential for competing genotypes in a co-infection.

Finally, pathogens can modify the host environment in diverse ways with consequences for neighbours locally and in distant niches within the host. For example, experimental murine infection revealed that *P. chabaudi* may diminish the barrier function of the intestinal wall resulting in enhanced translocation and dissemination of non-typhoidal *Salmonella* from the intestine. Correspondingly, *Salmonella* co-infection resulted in alterations in the immune response to *P. chabaudi* [78]. This is consistent with a pathological association between malaria and gastrointestinal disturbance involving non-typhoidal salmonella [79].

## 5. Evolutionary implications and areas that need development

In addition to field evaluations and experimental laboratory studies, insight into the dynamics of parasite co-infection benefit from modelling and theoretical approaches. Modelling can assist with the interpretation of spatial/geographical mapping to identify areas of high co-infection risk, as well as determining the strength of interactions between co-infecting parasites *in vivo* [80,81] and therapeutic impact [82]. Such data will become increasingly important in future, as climate change and human activities alter parasite and vector distributions, which may increase or reduce opportunities for species to interact. Single- and co-infection data from the field and multi-omic analyses in the laboratory can

also be extended through mathematical models to predict the effects of co-infections.

More analyses and theoretical input are needed on the evolutionary consequences of co-infection. Simplistically, theory predicts that co-infection will favour the selection of parasites with increased virulence or heightened transmission potential [83]. There are data, however, which suggests that intermediate virulence levels may be optimal, as a trade-off between transmission and persistence [84]. Experimental evidence shows that virulent strains can have a competitive advantage, which can alter the distribution of less competitive strains. For example, a study of mixed strain *P. chabaudi* infections revealed that more virulent parasite strains had a competitive advantage, so that mixed infections could favour selection of yet more virulent parasites [85,86]. However, in the case of another rodent malaria parasite, *P. yoelii*, virulence was not linked to increased competitive success [87], indicating that within-host competition does not always select for more virulent parasites, as trade-offs come in to play [88]. Ultimately, the infection success of even the most virulent strains and species is governed by the composition of the infecting community within a host. Statistical models have illustrated positive and negative effects of different species combinations, while other experiments, using trematode parasites of amphibian hosts, showed that increased species richness diminished the infection success of the most virulent parasite species [80,89]. Which traits are selected for during within-host competition will vary in different contexts, depending upon the diverse biology of the hosts and infectious organisms involved in the interaction. Further, short-term experimental studies may not fully reflect the selective consequences of long-term coexistence between pathogens in the field.

It is clear that species composition can alter the infection success within a host and at the population level. Hence, more careful analyses are required before public health-and veterinary-interventions are undertaken in the field, given that a targeted approach against a particular parasite may alter the distribution and infection success of another species in co-infection scenarios. For example, malaria and lymphatic filariasis (LF) are co-endemic and transmitted by the same mosquito vector. Incorporating data into a susceptible-infected model indicated that the introduction of LF into a population reduced the prevalence of malaria [13]. Given these data, the authors caution against targeted interventions against LF through mass-drug administration, which may have unintended perverse effects such as an increase in the R0 of malaria. The predicted effects of different strategies to control animal African trypanosomiasis (AAT) were also recently explored using modelling approaches. These data indicated that insecticidal treatment of cattle alone could eliminate *T. brucei* from the local population, but that it would have little effect on two other species, namely *T. congolense* and *T. vivax*, which are maintained in reservoirs such as wildlife and small ruminants [90].

Simplistically, theory predicts that two parasites, which compete directly or indirectly, are incapable of occupying the same niche indefinitely [91]. Hence, niche adaptation and speciation as a result of co-infection and the resultant effects on parasite interactions warrants exploration. One species may actively exclude another from a particular host niche, or a species may exhibit avoidance behaviours to promote their fitness. This may underlie the tissue compartmentalization of different African trypanosome species, with *T. brucei* exhibiting

tropism for the adipose tissue [92] and skin, whereas other African trypanosomes preferentially occupy other niches. The differential preference for red blood cell types may reflect a similar phenomenon in malaria species.

Finally, we have focused primarily on interactions between parasites during co-infection but the ability of a pathogen to colonize a host can be strongly influenced by the pre-existing community of non-pathogenic organisms within that host, adding complexity to the network of potential interactions. Although beyond what can be discussed here, the impact of the microbiome, for example, on the immune system of the hosts has clear relevance in the context of parasite infections in their hosts. This is true of malaria, where the gut microbiota influences disease severity caused by, and the immune response to, the parasite (e.g. [93–95]). A further particularly relevant example concerns trypanosome infections in the tsetse fly, where tsetse endosymbionts can have significant impact on the vectorial capacity of the arthropod [96]. The ability to manipulate both the microbiome in humans via probiotics [95] or tsetse endosymbionts through gene drive [97] or paratransgenesis [98] offers exciting possibility to control the pathology and transmission of each parasite.

# 6. Conclusion

In this review we have discussed, using trypanosome and *Plasmodium* parasites as exemplars, how an understanding of pathogen biology requires a broad analysis of the context of infection—particularly relating to competition and cooperation with other organisms. Traditionally, these interactions have been difficult to study comprehensively and pathogen research has focused on simple but unrepresentative one-host one-pathogen models. While informative and tractable, this inevitably omits the contribution of multiple other factors that can substantially impact the virulence and transmission of pathogens, and on the ability to control them immunologically or therapeutically. With the advent of high-resolution and high-throughput genomic and metagenomic surveillance and big data approaches to epidemiological, molecular and immunological study, multi-species analysis is now possible as is a dynamic study of interactions over time. Nonetheless, these methodological developments do not remove the value of well-controlled laboratory studies where complex interactions can be studied with limited variables. Fortunately, such experimental systems are now tractable for trypanosomes and

*Plasmodium* parasites in particular, with the availability of molecular reporters for different parasite stages of development, antigenic expression or the dynamic monitoring of niche occupation. This represents a particularly exciting platform where the rigour of laboratory study can be combined with a wide bandwidth of information input, helping to deconvolve the contributions and interactions of many different components in a pathogen infection with closer proximity to the real world. Inevitably such studies will remain incomplete and oversimplified but they will provide the structural framework on which relevant parameters can be identified or tested for impact. The field of co-infection biology is one that particularly lends itself to an interdisciplinary approach. By working together epidemiologists, molecular biologists, modellers and evolutionary biologists can build a picture of where and how co-infections are negatively impacting human and animal health. Most importantly, through the combination of quantitative information and evolutionary theory, the consequences of the deployment of therapies can be better understood to avoid the perverse impact of controlling one pathogen while increasing the virulence, transmission or pathology of others. Indeed, drug development often focuses on a target organism without consideration of the wider impact on human and animal health over time, where vacant niches become occupied by new threats, or there is selection for enhanced transmission or virulence in related and unrelated pathogens. Understanding the interactions between the community of organisms that contribute to pathogen impact in the field is necessary if control approaches and therapies are to be sustained, economical and safe.

Data accessibility. This article has no additional data.

Authors' contributions. F.V.: writing—original draft, writing—review and editing; K.R.M.: funding acquisition, writing—original draft, writing—review and editing; E.S.: writing—original draft, writing—review and editing. All authors gave final approval for publication and agreed to be held accountable for the work performed therein.

Competing interests. We declare we have no competing interests.

Funding. Work in K.R.M.'s laboratory is funded by the Wellcome Trust (grant nos. 221 717/Z/20/Z, 206815/Z/17/Z, 103740/Z/14/Z). F.V. is funded by a Wellcome Trust PhD studentship on the University of Edinburgh Wellcome Trust 'Hosts, Pathogens and Global Health' Programme (grant no. 108905/Z/15/Z).

Acknowledgements. We thank Dr Petra Schneider, University of Edinburgh, UK, for comments on the manuscript.

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
