## [Peer Review File · Proceedings of the Royal Society B: Biological Sciences]

Review History

RSPB-2021-0947.R0 (Original submission)

Review form: Reviewer 1

Recommendation

Reject – article is not of sufficient interest (we will consider a transfer to another journal)

Scientific importance: Is the manuscript an original and important contribution to its field?

Marginal

General interest: Is the paper of sufficient general interest?

Acceptable

Quality of the paper: Is the overall quality of the paper suitable?

Poor

Is the length of the paper justified?

Yes

Should the paper be seen by a specialist statistical reviewer?

No

Do you have any concerns about statistical analyses in this paper? If so, please specify them explicitly in your report.

No

It is a condition of publication that authors make their supporting data, code and materials available - either as supplementary material or hosted in an external repository. Please rate, if applicable, the supporting data on the following criteria.

Is it accessible?

N/A

Is it clear?

N/A

Is it adequate?

N/A

Do you have any ethical concerns with this paper?

No

Comments to the Author

This study provides a perspective on co-infection by discussing some of the ecological and molecular components that are less frequently addressed, including pros and cons of laboratory and field work. The manuscript includes some interesting examples however, I found the work lacking of conceptualization, theoretical background and critical discussion, and rather focused on listing examples. I also found difficult to follow the structure of the work as it jumped across themes without clear connections, sometimes from disparate topics and again, without critical discussion.

It would have been helpful to examine just few key topics, address them from the many angles the authors are interested in, provide some strong discussion and make the perspective more compelling.

For the section on within-host parasite's niche selection during co-infection I highly recommend to read some of exceptional work done in 1961 and 1962 by Holmes or Stock and Holmes 1988.

Review form: Reviewer 2

Recommendation

Reject – article is not of sufficient interest (we will consider a transfer to another journal)

Scientific importance: Is the manuscript an original and important contribution to its field?

Good

General interest: Is the paper of sufficient general interest?

Marginal

Quality of the paper: Is the overall quality of the paper suitable?

Good

Is the length of the paper justified?

Yes

Should the paper be seen by a specialist statistical reviewer?

No

Do you have any concerns about statistical analyses in this paper? If so, please specify them explicitly in your report.

No

It is a condition of publication that authors make their supporting data, code and materials available - either as supplementary material or hosted in an external repository. Please rate, if applicable, the supporting data on the following criteria.

Is it accessible?

N/A

Is it clear?

N/A

Is it adequate?

N/A

Do you have any ethical concerns with this paper?

No

Comments to the Author

Overall a well written and comprehensive review, this manuscript was clear and easy to read, and should be of great interest to disease ecologists and parasitologists. The coverage of trypanosome coinfections was especially interesting, as this is an area that doesn't receive much attention (unlike malaria coinfections).

However, for a general reader, the manuscript reads as a long list of examples and would benefit from being drawn together into a clearer conceptual narrative. Ideally, readers not presently working on coinfection would emerge with a sense of how to organize their understanding of coinfections, rather than with a sense of overwhelming complexity. And/or, if the authors were able to write this from a focused applied perspective (management of trypanosomiasis, perhaps?), we suspect that it would increase the general interest of the paper.

Thus our major suggestion for revision for a general journal like Proc B is this: please find ways to make this more appealing to a wide audience of biologists. Perhaps offer more general rules or guiding principles for understanding the outcomes of coinfections, for example, when multiple of the mechanisms depicted in figure 2 are acting/interacting? Another idea might be to frame the whole thing around guiding principles that make sense of the tryp/malaria examples, with other systems only mentioned in passing. The content was best when focused on trypanosomes and malaria, while most of the other examples seemed to come a bit at random.

A few other suggestions are below.

- 1) The introduction needs some references added - at the moment there are no citations at all (pages 3 - 4).
- 2) First line of page 6 - is it possible to briefly summarise these "striking effects"?
- 3) First sentence of last paragraph on page 7: "Alternatively, coinfection can result in reduced virulence where parasites with low relatedness directly interfere with each other." Is this only for parasites with low relatedness? Later in the paragraph you talk about interactions between two trypanosomes, which at some scales would be considered closely related. Maybe define what you mean by "low relatedness"?
- 4) Since you don't address immune-interactions in the review, we would suggest noting this in the abstract and/or introduction - maybe "with a focus on direct, non-immune mediated interactions..." or something similar.

Decision letter (RSPB-2021-0947.R0)

01-Jun-2021

Dear Professor Matthews:

I am writing to inform you that your manuscript RSPB-2021-0947 entitled "Parasite coinfection: an ecological, molecular and experimental perspective" has, in its current form, been rejected for publication in Proceedings B.

This action has been taken on the advice of referees, who have recommended that substantial revisions are necessary. The reviewers are fairly unanimous in their criticism and raise a number of important points, including the observation that the current version feels more like a set of (interesting) examples than a narrative. With this in mind we would be happy to consider a resubmission of the perspective, provided the comments of the referees are fully addressed. However please note that this is not a provisional acceptance.

Sincerely,
Professor Hans Heesterbeek
<mailto:proceedingsb@royalsociety.org>

Reviewer(s)' Comments to Author:

Referee: 1

Comments to the Author(s)

This study provides a perspective on co-infection by discussing some of the ecological and molecular components that are less frequently addressed, including pros and cons of laboratory and field work. The manuscript includes some interesting examples however, I found the work lacking of conceptualization, theoretical background and critical discussion, and rather focused on listing examples. I also found difficult to follow the structure of the work as it jumped across themes without clear connections, sometimes from disparate topics and again, without critical discussion.

It would have been helpful to examine just few key topics, address them from the many angles the authors are interested in, provide some strong discussion and make the perspective more compelling.

For the section on within-host parasite's niche selection during co-infection I highly recommend to read some of exceptional work done in 1961 and 1962 by Holmes or Stock and Holmes 1988.

Referee: 2

Comments to the Author(s)

Overall a well written and comprehensive review, this manuscript was clear and easy to read, and should be of great interest to disease ecologists and parasitologists. The coverage of trypanosome coinfections was especially interesting, as this is an area that doesn't receive much attention (unlike malaria coinfections).

However, for a general reader, the manuscript reads as a long list of examples and would benefit from being drawn together into a clearer conceptual narrative. Ideally, readers not presently working on coinfection would emerge with a sense of how to organize their understanding of coinfections, rather than with a sense of overwhelming complexity. And/or, if the authors were able to write this from a focused applied perspective (management of trypanosomiasis, perhaps?), we suspect that it would increase the general interest of the paper.

Thus our major suggestion for revision for a general journal like Proc B is this: please find ways to make this more appealing to a wide audience of biologists. Perhaps offer more general rules or guiding principles for understanding the outcomes of coinfections, for example, when multiple of the mechanisms depicted in figure 2 are acting/interacting? Another idea might be to frame the whole thing around guiding principles that make sense of the tryp/ malaria examples, with other systems only mentioned in passing. The content was best when focused on trypanosomes and malaria, while most of the other examples seemed to come a bit at random.

A few other suggestions are below.

- 1) The introduction needs some references added - at the moment there are no citations at all (pages 3 - 4).
- 2) First line of page 6 - is it possible to briefly summarise these "striking effects"?
- 3) First sentence of last paragraph on page 7: "Alternatively, coinfection can result in reduced virulence where parasites with low relatedness directly interfere with each other." Is this only for parasites with low relatedness? Later in the paragraph you talk about interactions between two trypanosomes, which at some scales would be considered closely related. Maybe define what you mean by "low relatedness"?
- 4) Since you don't address immune-interactions in the review, we would suggest noting this in the abstract and/or introduction - maybe "with a focus on direct, non-immune mediated interactions..." or something similar.

Author's Response to Decision Letter for (RSPB-2021-0947.R0)

See Appendix A.

RSPB-2021-2155.R0

Review form: Reviewer 1

Recommendation

Reject - article is scientifically unsound

Scientific importance: Is the manuscript an original and important contribution to its field?
Marginal

General interest: Is the paper of sufficient general interest?
Good

Quality of the paper: Is the overall quality of the paper suitable?
Marginal

Is the length of the paper justified?
Yes

Should the paper be seen by a specialist statistical reviewer?
No

Do you have any concerns about statistical analyses in this paper? If so, please specify them explicitly in your report.
No

It is a condition of publication that authors make their supporting data, code and materials available - either as supplementary material or hosted in an external repository. Please rate, if applicable, the supporting data on the following criteria.

Is it accessible?
N/A

Is it clear?
N/A

Is it adequate?
N/A

Do you have any ethical concerns with this paper?
No

Comments to the Author

I read this perspective before and while some aspects have been clarified, I still think that the paper needs some careful revision.

The initial argument about field and lab studies is confusing at time, and the comment that disentangling co-infections in simple laboratory studies are almost impossible (line 67) is in contradiction with what is said in the rest of the paper.

There are many new sections that are all extremely interesting and important in the context of co-infection, however, there is a general tendency to briefly touch on many issues but not to address any in detail, and I felt left half-way begging for a comment or a discussion of those patterns. If included, some of the comments are quite generic or do not add much to what we already know about co-infections. I like section 4, probably because I don't not know much about it but I found the other sections missing the Authors' perspective.

There are parts that still tend to be a list of examples and there is no clear sequential logic with some of the sections. It would be probably better just to focus on few key issues and develop a compelling discussion around them.

Decision letter (RSPB-2021-2155.R0)

18-Oct-2021

Dear Professor Matthews:

Your revised manuscript for an invited Perspective has now been peer reviewed and the review has been assessed by me. The reviewer's comments (not including confidential comments to the Editor) are included at the end of this email for your reference. Both the reviewer and I think that the focus on two parasite systems has substantially improved the manuscript. As you will see, the reviewer would prefer more in-depth explanatory coverage of a limited number of aspects, instead of briefly touching on a large number of relevant dimensions. I certainly sympathize with that view, but also see that it is difficult to find an optimal balance, if that even exists. As a Perspective, the manuscript does provide a large group of interested but non-expert readers with a birds-eye view of the area and a guide to some of the pertinent literature. While not expecting a complete overhaul of the current version, I would like to invite you though to try and revise your manuscript in light of the overall comment by the reviewer.

Research ethics:

Use of animals and field studies:

It is a condition of publication that you make available the data and research materials supporting the results in the article (<https://royalsociety.org/journals/authors/author-guidelines/#data>). Datasets should be deposited in an appropriate publicly available repository and details of the associated accession number, link or DOI to the datasets must be included in the Data Accessibility section of the article (<https://royalsociety.org/journals/ethics-policies/data-sharing-mining/>). Reference(s) to datasets should also be included in the reference list of the article with DOIs (where available).

If you wish to submit your data to Dryad (<http://datadryad.org/>) and have not already done so you can submit your data via this link [http://datadryad.org/submit?journalID=RSPB&manu=\(Document not available\)](http://datadryad.org/submit?journalID=RSPB&manu=(Document%20not%20available)), which will take you to your unique entry in the Dryad repository.

Please submit a copy of your revised paper within three weeks. If we do not hear from you within this time your manuscript will be rejected. If you are unable to meet this deadline please let us know as soon as possible, as we may be able to grant a short extension.

Best wishes,
Professor Hans Heesterbeek
<mailto:proceedingsb@royalsociety.org>

Reviewer(s)¹ Comments to Author:

Referee: 1

Comments to the Author(s).

I read this perspective before and while some aspects have been clarified, I still think that the paper needs some careful revision.

The initial argument about field and lab studies is confusing at time, and the comment that disentangling co-infections in simple laboratory studies are almost impossible (line 67) is in contradiction with what is said in the rest of the paper.

There are many new sections that are all extremely interesting and important in the context of co-infection, however, there is a general tendency to briefly touch on many issues but not to address any in detail, and I felt left half-way begging for a comment or a discussion of those patterns. If included, some of the comments are quite generic or do not add much to what we already know about co-infections. I like section 4, probably because I don't not know much about it but I found the other sections missing the Authors' perspective.

There are parts that still tend to be a list of examples and there is no clear sequential logic with some of the sections. It would be probably better just to focus on few key issues and develop a compelling discussion around them.

Author's Response to Decision Letter for (RSPB-2021-2155.R0)

See Appendix B.

RSPB-2021-2155.R1

Review form: Reviewer 2

Recommendation

Accept with minor revision (please list in comments)

Scientific importance: Is the manuscript an original and important contribution to its field?

Excellent

General interest: Is the paper of sufficient general interest?

Excellent

Quality of the paper: Is the overall quality of the paper suitable?

Excellent

Is the length of the paper justified?

Yes

Should the paper be seen by a specialist statistical reviewer?

No

Do you have any concerns about statistical analyses in this paper? If so, please specify them explicitly in your report.

No

It is a condition of publication that authors make their supporting data, code and materials available - either as supplementary material or hosted in an external repository. Please rate, if applicable, the supporting data on the following criteria.

Is it accessible?

N/A

Is it clear?

N/A

Is it adequate?

N/A

Do you have any ethical concerns with this paper?

No

Comments to the Author

Having reviewed the original MS but having missed the early autumn round of review (i.e., my senior graduate student and I were the original "referee #2" but then I somehow missed any invitation to review the first revision), we were delighted to see that the authors followed our suggestion to reframe the review around the trypanosome and malaria systems. This refined focus works really well, as it highlights the best and most exciting examples. We agree with the authors' suggestion that the focus also makes the review very timely. Furthermore, we find that the restructuring of the arguments throughout the MS has made the review both clearer and more compelling. We just have a handful of final typo-scale corrections to suggest (see below). Excellent work!

Line 155 – citation missing?

Line 196 – it seems as though you only discuss your second definition of virulence in the following section (which is arguably the more common definition, anyway, in co-infection studies). Perhaps remove the first definition ("overall parasite number") from this sentence, or else clarify that you are only including it to highlight that it's not the definition you are using.

Line 303 – maybe spell out the controversy?

Line 311 - citation missing?

Line 478 – suggest replacing the comma before "by working together" with an em-dash, or just start a new sentence there

Decision letter (RSPB-2021-2155.R1)

08-Dec-2021

Dear Professor Matthews

I am pleased to inform you that your manuscript RSPB-2021-2155.R1 entitled "Parasite coinfection: an ecological, molecular and experimental perspective" has been accepted for publication in Proceedings B.

The referee has recommended publication, but also suggests some minor revisions to your manuscript. Therefore, I invite you to respond to the comments and revise your manuscript. Because the schedule for publication is very tight, it is a condition of publication that you submit the revised version of your manuscript within 7 days. If you do not think you will be able to meet this date please let us know.

When submitting your revised manuscript, you will be able to respond to the comments made by the referee(s) and upload a file "Response to Referees". You can use this to document any changes

you make to the original manuscript. We require a copy of the manuscript with revisions made since the previous version marked as 'tracked changes' to be included in the 'response to referees' document.

Sincerely,
 Professor Hans Heesterbeek
 Editor, Proceedings B
<mailto:proceedingsb@royalsociety.org>

Reviewer(s)' Comments to Author:

Referee: 2

Comments to the Author(s)

Having reviewed the original MS but having missed the early autumn round of review (i.e., my senior graduate student and I were the original "referee #2" but then I somehow missed any invitation to review the first revision), we were delighted to see that the authors followed our suggestion to reframe the review around the trypanosome and malaria systems. This refined focus works really well, as it highlights the best and most exciting examples. We agree with the authors' suggestion that the focus also makes the review very timely. Furthermore, we find that the restructuring of the arguments throughout the MS has made the review both clearer and more compelling. We just have a handful of final typo-scale corrections to suggest (see below). Excellent work!

Line 155 – citation missing?

Line 196 – it seems as though you only discuss your second definition of virulence in the following section (which is arguably the more common definition, anyway, in co-infection studies). Perhaps remove the first definition ("overall parasite number") from this sentence, or else clarify that you are only including it to highlight that it's not the definition you are using.

Line 303 – maybe spell out the controversy?

Line 311 - citation missing?

Line 478 – suggest replacing the comma before "by working together" with an em-dash, or just start a new sentence there

Author's Response to Decision Letter for (RSPB-2021-2155.R1)

See Appendix C.

Decision letter (RSPB-2021-2155.R2)

09-Dec-2021

Dear Professor Matthews

I am pleased to inform you that your manuscript entitled "Parasite coinfection: an ecological, molecular and experimental perspective" has been accepted for publication in Proceedings B.

You can expect to receive a proof of your article from our Production office in due course, please check your spam filter if you do not receive it. PLEASE NOTE: you will be given the exact page

length of your paper which may be different from the estimation from Editorial and you may be asked to reduce your paper if it goes over the 10 page limit.

Data Accessibility section

Open Access

Paper charges

Sincerely,

Appendix A

SCHOOL of BIOLOGICAL SCIENCES
Institute of Immunology and Infection Research

Professor Keith R. Matthews, PhD FRSE FMedSci FRS
Dean of Bioscience Partnerships
School of Biological Sciences,
University of Edinburgh
Edinburgh, EH9 3JT
United Kingdom

Tel: +441316513639
Fax: +441316513670

keith.matthews@ed.ac.uk

The University of Edinburgh
Ashworth Laboratories
The King's Buildings
West Mains Road
Edinburgh EH9 3JT

Direct Dial 0131 650

Switchboard 0131 650 1000

Fax 0131 650 6564

28th September 2021

Thank you for sending on the reviewers' comments on our '*Perspective*' article. As they reflect, we had grappled with how best to cover the complexity of this topic and appreciate their suggestions for how we might improve the accessibility and coherence of the review. On that basis, we have adopted their suggestions to examine a more specific area of the broad overall topic and set the review in that context. Particularly we have followed Referee 2s suggestion to concentrate on two parasite systems, trypanosomes and Plasmodium, which we feel was beneficial as research on both parasites has an increasing recent attention on multi-strain and multispecies coinfections. We believe this will make the review more timely. Also, we have overhauled the structure of the review to hopefully improve its clarity and the accessibility of the key concepts in the topic. We have also removed many of the diverse examples within the review to hopefully avoid a 'list-like' perception, focusing instead on recent publications from the trypanosome and Plasmodium literature to make the key conceptual points.

The article has been extensively revised, and a great deal of text has been removed and the overall structure and narrative has evolved significantly. We hope it is now considered a more valuable and accessible coverage of this fascinating topic.

Kind regards

Keith

Professor Keith Matthews,
University of Edinburgh

Reviewer(s)' Comments to Author:

Referee: 1

Comments to the Author(s)

This study provides a perspective on co-infection by discussing some of the ecological and molecular components that are less frequently addressed, including pros and cons of laboratory and field work. The manuscript includes some interesting examples however, I found the work lacking of conceptualization, theoretical background and critical discussion, and rather focused on listing examples. I also found difficult to follow the structure of the work as it jumped across themes without clear connections, sometimes from disparate topics and again, without critical discussion.

It would have been helpful to examine just few key topics, address them from the many angles the authors are interested in, provide some strong discussion and make the perspective more compelling.

For the section on within-host parasite's niche selection during co-infection I highly recommend to read some of exceptional work done in 1961 and 1962 by Holmes or Stock and Holmes 1988.

We have clarified the structure of the review and broken it down into subsections, with each addressed using examples from the trypanosome and Plasmodium systems. A great deal of text and unrelated examples have been removed or revised.

Referee: 2

Comments to the Author(s)

Overall a well written and comprehensive review, this manuscript was clear and easy to read, and should be of great interest to disease ecologists and parasitologists. The coverage of trypanosome coinfections was especially interesting, as this is an area that doesn't receive much attention (unlike malaria coinfections).

However, for a general reader, the manuscript reads as a long list of examples and would benefit from being drawn together into a clearer conceptual narrative. Ideally, readers not presently working on coinfection would emerge with a sense of how to organize their understanding of coinfections, rather than with a sense of overwhelming complexity. And/or, if the authors were able to write this from a focused applied perspective (management of trypanosomiasis, perhaps?), we suspect that it would increase the general interest of the paper.

Thus our major suggestion for revision for a general journal like Proc B is this: please find ways to make this more appealing to a wide audience of biologists. Perhaps offer more general rules or guiding principles for understanding the outcomes of coinfections, for example, when multiple of the mechanisms depicted in figure 2 are acting/interacting? Another idea might be to frame the whole thing around guiding principles that make sense of the tryp/malaria examples, with other systems only mentioned in passing. The content was best when focused on trypanosomes and malaria, while most of the other examples seemed to come a bit at random.

We have taken the Referee's proposal to frame the article around trypanosomes and malaria- thank you for the suggestion. We have extensively revised the content and structure and focused on examples using these two parasites- that serve nicely to highlight many major concepts in coinfection. We also think the current research on these parasites and methodological developments makes the review particularly timely. We hope this will increase its access and relevance.

A few other suggestions are below.

1) The introduction needs some references added - at the moment there are no citations at all (pages 3 – 4).

Now added

2) First line of page 6 - is it possible to briefly summarise these "striking effects"?

This section is now deleted.

3) First sentence of last paragraph on page 7: "Alternatively, coinfection can result in reduced virulence where parasites with low relatedness directly interfere with each other." Is this only for parasites with low relatedness?

Later in the paragraph you talk about interactions between two trypanosomes, which at some scales would be considered closely related. Maybe define what you mean by “low relatedness”?

We agree there is ambiguity in the extent of relatedness, which in the cited article discusses relative relatedness. In some ways it depends on how selection operates and so is a theoretical concept in general terms. In the text, we highlight a couple of examples where the parasites are less related (different trypanosome species) or more related (different strains of the same species but with different virulence characteristics). We now include speculation that it is not necessarily overall relatedness but potentially the relatedness at key virulence loci that determines the overall interference between coinfecting parasites. The modified text is line 179-181.

4) Since you don't address immune-interactions in the review, we would suggest noting this in the abstract and/or introduction – maybe “with a focus on direct, non-immune mediated interactions...” or something similar.

Now added

Journal Name: Proceedings of the Royal Society B

Journal Code: RSPB

Print ISSN: 0962-8452

Online ISSN: 1471-2954

Journal Admin Email: proceedingsb@royalsociety.org

MS Reference Number: RSPB-2021-0947

Article Status: REJECTED

MS Dryad ID: RSPB-2021-0947

MS Title: Parasite coinfection: an ecological, molecular and experimental perspective

MS Authors: Venter, Frank; Matthews, Keith; Silvester, Eleanor

Contact Author: Keith Matthews

Contact Author Email: keith.matthews@ed.ac.uk

Contact Author Address 1: School of Biological Sciences

Contact Author Address 2: West Mains Road

Contact Author Address 3:

Contact Author City: Edinburgh

Contact Author State:

Contact Author Country: United Kingdom of Great Britain and Northern Ireland

Contact Author ZIP/Postal Code: EH9 3JT

Keywords: Parasite, coinfection, Trypanosoma, Plasmodium

Abstract: Laboratory studies of pathogens aim to limit complexity in order to disentangle the important parameters contributing to an infection. However, pathogens rarely exist in isolation, and hosts may sustain coinfections with multiple disease agents. These interact with each other and with the host immune system dynamically, with disease outcomes affected by the composition of the community of infecting pathogens, their order of colonisation, competition for niches and nutrients and immune modulation. Here we review how ecological and experimental studies have revealed the interactions between pathogens in mammal hosts and arthropod vectors and, focusing on trypanosomes and malaria parasites, discuss recently developed laboratory models for coinfection. The implication of pathogen coinfection for disease therapy is also discussed.

EndDryadContent

Appendix B

Keith R. Matthews, FRSE FMedSci FRS

Professor of Parasite Biology and
Dean, Bioscience Partnerships
School of Biological Sciences,
University of Edinburgh
Edinburgh, EH9 3JT

Tel: +441316513639

Email: keith.matthews@ed.ac.uk

1st November 2021

SCHOOL of BIOLOGICAL SCIENCES
Institute of Immunology and Infection Research

The University of Edinburgh
Ashworth Laboratories
The King's Buildings
West Mains Road
Edinburgh EH9 3JT

Direct Dial 0131 650

Switchboard 0131 650 1000

Fax 0131 650 6564

Response to referee comments

Thank you for sending on the decision on our perspective article. As you highlight in your decision letter, our Perspective covers a broad topic. In our first revision, as was requested, we very significantly revised our submission and narrowed it down to consider just two parasite groups (Trypanosomes and malaria) which are particularly suited to consideration of coinfections, given the prevalence of epidemiological and field data and the availability of tractable experimental systems. After our revision the referee has asked for yet more focus and detail but, as you reflect in your decision letter, we struggle to accommodate this request whilst also retaining the broad perspective that we think is particularly valuable at present. This is because researchers are only now beginning to realise the importance of coinfections for these parasites and have only recently had available the tools to tease apart the interactions. This is why we believe an overview considering many aspects of the topic (encompassing epidemiological, experimental, mechanistic and modelling approaches) is most appropriate at this time. We agree with your view that combining an overview perspective and going into significant depth for particular topics is essentially an impossible task.

Nonetheless we have tried where possible to introduce additional comments and discussion around the different parts of the Perspective to improve its flow and readability, or to emphasise some key points. These changes are annotated on the 'track changes' copy of the manuscript uploaded with this rebuttal. We do not think it possible to fully address the request for more detail and opinion while also retaining the comprehensive overview of this enormous subject. Instead, by setting out the important considerations, limitations and excitement of studying and understanding coinfection we hope we will increase the recognition of this important component of pathogen-pathogen interactions and stimulate further detailed work. Future work and other reviews will focus on the details of different components of coinfection biology and in different systems. However, at this time and at this stage we think it more valuable to highlight the suite of approaches available and how they can be exploited using an interdisciplinary perspective to understand how pathogens interact outside the extremes of one-host-one pathogen laboratory models or population scale epidemiological studies.

I hope you will now consider the perspective article acceptable for publication.

Kind regards

Professor Keith R. Matthews FRSE FMedSci FRS

Appendix C

Response to referees

We were very pleased the referee was so positive about our revised submission. We believe we have addressed all of their suggestions for revision –although the line numbers cited did not correspond to the submitted version. Nonetheless we think we have understood their intent.

Keith Matthews